# Tumor Treating Fields Combine with Temozolomide for Newly Diagnosed Glioblastoma: A Retrospective Analysis of Chinese Patients in a Single Center

**DOI:** 10.3390/jcm11195855

**Published:** 2022-10-03

**Authors:** Chunjui Chen, Hao Xu, Kun Song, Yi Zhang, Junyan Zhang, Yang Wang, Xiaofang Sheng, Lingchao Chen, Zhiyong Qin

**Affiliations:** 1Department of Neurosurgery, Huashan Hospital, Fudan University, Shanghai 200040, China; 2National Center for Neurological Disorders, Shanghai 200040, China; 3Shanghai Key Laboratory of Brain Function Restoration and Neural Regeneration, Shanghai 200040, China; 4Neurosurgical Institute of Fudan University, Shanghai 200040, China; 5Shanghai Clinical Medical Center of Neurosurgery, Shanghai 200040, China; 6Bothwin Clinical Study Consultant, Shanghai 201702, China; 7Branch of Clinical Epidemiology and Evidence-Based Medicine, Shanghai Medical Association, Shanghai 200040, China; 8Department of Radiation Oncology, Huashan Hospital, Fudan University, Shanghai 200040, China

**Keywords:** glioblastoma, tumor treating fields, chemotherapy, retrospective cohort

## Abstract

Introduction: TTFields plus Temozolomide (TTFields/TMZ) extended survival versus TMZ alone in newly diagnosed glioblastoma (GBM) patients in the EF-14 trial. We have reported a retrospective analysis of newly diagnosed Chinese GBM patients who received TTFields/TMZ treatment and TMZ treatment from August 2018 to May 2021 in Huashan hospital in Shanghai. Methods: Overall survival (OS) and progression-free survival (PFS) curves were constructed using the Kaplan–Meier method. A Cox proportional hazards regression model, propensity score matched data, and inverse probability of treatment weighting (IPTW) based on propensity score were used to assess the effect of TTFields and account for confounding factors. Results: In the preliminary analysis, the median PFS in TTFields/TMZ group was 16 months (95% CI, 9.6–24.6) versus 11 months (95% CI, 9–12) in TMZ group (*p* < 0.05). Median overall survival was 21.8 months (95% CI, 17.4-NA) with TTFields/TMZ versus 15 months (HR = 0.43; 95% CI, 13–18) with TMZ alone. The multivariate analysis identified surgery type, STUPP scheme, IDH status, and TTFields use as favorable prognostic factors. After PSM adjustment, the variate among the groups was similar, except that the methylation rate of MGMT promoter remained high in the TMZ group (12 v 32 months; *p* = 0.011). Upon IPTW Survival analysis, TTFields was associated with a significantly lower risk of death (HR = 0.19 in OS; 95% CI, 0.09–0.41) and progression (HR = 0.35; 95% CI 0.14–0.9) compared with TMZ group. Conclusion: In the final analysis of our single-center Chinese patients with glioblastoma, adding TTFields to temozolomide chemotherapy resulted in statistically significant improvement in PFS and OS.

## 1. Introduction

Adult glioblastoma is one of the most fatal and challenging diseases and is associated with repeated recurrence and inferior prognosis [1,2,3]. Multimodal therapy of glioblastoma includes surgery, radiotherapy, systemic chemotherapy, and target therapy, which have been proven to result in limited improvement [4]. Most clinical trials revealed that the overall survival was around 15 months [5,6,7], and 5-year survival rate was only 5.8% [8,9,10]. Second-line therapies such as lomustine, carmustine, and temozolomide rechallenging after standard concurrent chemoradiation and adjuvant chemotherapy might only benefit a subgroup of patients with MGMT promoter methylation [11,12].

Tumor treating fields (TTFields) have changed the first-line clinical practice of glioblastoma worldwide by its promising therapeutic efficacy [13,14,15]. Delivering low-intensity, intermediate-frequency of 200 kHz alternating electric fields, TTFields present with the inhibitory effect of tumor cell proliferation by interfering with mitotic spindle formation during metaphase [16,17,18].

An EF-14 randomized trial revealed a median progression-free survival (PFS) of 6.7 months in the TTFields plus temozolomide group and PFS of 4 months in the temozolomide-alone group. The median overall survival (OS) of TTFields plus temozolomide group was 20.9 months, and OS of temozolomide-alone group was 16.0 months [14,19]. This represented the first positive outcome of all the therapies in newly diagnosed glioblastoma since Stupp protocols in 2005 [15]; however, it was devoid of clinical improvements in recurrent glioblastoma [20].

Correlations between genetic alternations and therapeutic efficacy have also been investigated in patients treated with TTFields [21]. There were undergoing trials of combinations of regiments as well, which were intended to facilitate the interaction between TTFields and other medical prescriptions [22].

This article is the first report on TTFields for the Chinese population. Our objective of this retrospective study was to evaluate the efficacy and safety of TTFields combined with TMZ during maintenance therapy versus TMZ alone in Chinese patients.

## 2. Methods and Materials

### 2.1. Patients and Tumor Samples

We included glioblastoma patients who underwent surgery followed by Stupp regimen and TTFields treatment at the Department of Neurosurgery, Huashan Hospital, Fudan University, between August 2018 and February 2021. Patients meeting the following criteria were eligible for the TTFields group: (1) age 18 or older; (2) histological diagnosis of GBM; (3) TTFields treatment for no less than 4 weeks; and (4) tumor available for genetic detection. Patients in the Non-TTFields group underwent surgery at Huashan Hospital between January 2016 and October 2017 and were required to meet all the above criteria, excluding criterion 3. The ethics committee approved this study of Huashan Hospital, Fudan University. Informed consent was obtained from all patients. Approval was granted by the Ethics Committee of Huashan Hospital, Fudan University Shanghai, China 200040 (No. KY2015-256).

Baseline characteristics included age, sex, Karnofsky performance status score (KPS), the extent of resection, tumor location, TTFields usage, progression-free survival (PFS), and overall survival (OS). The extent of resection was classified as gross total resection (GTR), partial resection, and biopsy. GTR was determined based on both surgical findings and postoperative images. Complete resection was achieved during surgery and confirmed by postoperative T1-weighted magnetic resonance imaging with contrast (no residual enhancement was observed on MR imaging). Patient compliance was assessed monthly as the average percentage of each day the TTFields treatment was conducted (out of each 24 h period).

### 2.2. Molecular Analysis

Glioma samples were obtained during surgical resection, snap-frozen by liquid nitrogen, and stored at −80 ℃. DNA of eligible quality and quantity was extracted from brain tumor tissue following the manufacturer’s instructions (BlackPREP FFPE kit, Analytik, Jena, Germany). The mutation status of IDH and telomerase reverse transcriptase (TERT) promoter were determined by Sanger sequencing (HITACHI 3500xL Dx Genetic Analyzer, Applied Biosystems Inc., Waltham, MA, USA). MGMT promoter methylation was detected by pyrosequencing (Pyromark Q24 instrument, Hildesheim, Germany).

### 2.3. Statistical Analysis

Continuous variables with normal distribution were presented as the mean followed by standard deviation. The abnormal distribution of continuous variables was introduced as the median, followed by the interquartile range from the first quartile to the third quartile (Q1–Q3). Student’s t-test and Wilcoxon rank-sum test were used for normally and abnormally distributed quantitative data at baseline. Categorical variables at baseline were analyzed by chi-squared or Fisher’s exact test.

Propensity score matching (PSM) was performed to match patients of the TTFields group with the Non-TTFields group. The propensity score data set was constructed using the multivariable logistic regression model, including age, sex, baseline KPS, the extent of resection, tumor location, IDH mutation status, MGMT promoter methylation, TERT promoter mutation status, and Stupp regimen. We used caliper matching with the caliper 0.2 of the pooled standard deviation of the logit of the propensity score. Patients in the TTFields group were matched 1:2 to patients in the Non-TTFields group.

The propensity data set generated the inverse probability of treatment weighting (IPTW) dataset. To balance those observable characteristics, each patient was weighted by the inverse probability of being in the TTFields group to compare to the Non-TTFields group.

For the primary endpoint, multivariate Cox regression assessed the association between overall survival and the two treatment groups. The model was adjusted by those variates whose *p*-value was no more than 0.10. Those variates with clinical meanings related to the primary endpoint were adjusted, ignoring the *p*-value. We analyzed the PSM dataset and the IPTW dataset with Cox regression for sensitivity analysis. The results were expressed as adjusted hazard ratios (HRs) with 95% confidence intervals (95% CIs). For the median time comparison, log-rank tests were used between the two groups. All hypothesis tests were two-sided, and values of <0.05 were considered statistically significant.

Stata SE 13 (Serial number 401306302851), R software version 4.2.0 (http://cran.r-project.org, accessed on 1 May 2022, and easy-R (www.empowerstats.com, accessed on 1 May 2022) were used for statistical analysis. GraphPad was used to generate figures.

## 3. Results

### 3.1. Study Population

Two hundred and sixty-seven patients were enrolled from 2019 to 2021 in our studies, and sixty-three patients received TTFields. The median age of the TTFields group was 51 and for Non-TTFields, 54. Around 48% were male patients, and 65% were in the non-TTFields group; the mean Karnofsky performance status scale score for each group was 80 and 90, respectively. In the TTFields group, 70% of gross total resection operations were achieved, versus 79% for Non-TTFields.

Tumor locations were classified into three categories: frontal lobe, superficial surface excluding frontal lobe, and midline/deep structure/infratentorial. The distribution of tumor location is shown in Table 1. All histological tissue slides were available, and an experienced pathologist confirmed the diagnosis of glioblastoma. MGMT methylation, TERT promoter, and IDH mutant status were tested in our studies. Standard Stupp protocol was performed in all patients in the TTFields group and reached 83% in the non-TTFields group.

Lastly, median compliance in the TTFields group was 87% during the recommended therapy period.

### 3.2. Prognostic Factors and Survival Rate

The median overall survival (OS) of the TTFields and non-TTFields group was 21.8 (95% CI 17.4–NA) versus 15 months (95% CI 13–18), for a proportional hazard ratio (HR) of 0.43 (95% CI 0.38–0.67, *p* < 0.001). The median progression-free survival of both groups was 16 (95% CI 9.6–24.6) versus 11 months (95% CI 9–12) with a proportional HR of 0.49 (95% CI 0.33–0.73, *p* < 0.001), as shown in Figure 1.

According to the univariate analysis of PFS and OS, as revealed in Table 2 and Table 3, administration of TTFields was a crucial prognostic factor in PFS and OS. The proportional HR was 0.53 (95% CI 0.37–0.75, *p* < 0.001) for PFS and was 0.40 (95% CI 0.27–0.60, *p* < 0.001) for OS.

Furthermore, clinical significances were discovered in the following indicators. Females demonstrated longer PFS and OS (HR = 0.71; 95% CI, 0.55–093; *p* = 0.013 and HR = 0.75; 95% CI, 0.57–0.98; *p* = 0.034, respectively). Gross total resection revealed superior prognosis than subtotal resection (HR = 2.17; 95% CI, 1.58–2.99; *p* < 0.001 and HR = 1.61; 95% CI, 1.17–2.22; *p* = 0.003, in PFS and OS, respectively). Biomarker such as IDH mutant status was only positive in PFS (HR = 0.58; 95% CI, 0.37–092; *p* = 0.013), as MGMT methylation status (HR = 1.54; 95% CI, 1.09–2.18; *p* = 0.014 and HR = 1.46; 95% CI, 1.02–2.08; *p* = 0.036, in PFS and OS, respectively) and TERT promoter status (HR = 1.58; 95% CI, 1.13–2.21; *p* = 0.007 and HR =1.65; 95% CI, 1.17–2.32; *p* = 0.006, in PFS and OS, respectively) were both related to outcome with significance. Lastly, accomplishment of Stupp protocol was also an impact factor (HR = 2.19; 95% CI, 1.51–3.17; *p* < 0.001 and HR =3.08; 95% CI, 2.21–4.47; *p* < 0.001, in PFS and OS, respectively).

### 3.3. Multivariate Analysis

Multivariate analysis by Cox regression (Table 2 and Table 3) demonstrated that the usage of TTFields prolonged the survival time (HR = 0.49; 95% CI, 0.33–0.73; *p* < 0.001 and HR = 0.43; 95% CI, 0.28–0.67; *p* < 0.001, in PFS and OS, respectively). A similar result was also discovered in the Stupp protocol accomplished subgroup (HR = 2.19; 95% CI, 1.51–3.17; *p* < 0.001 and HR = 3.08; 95% CI, 2.12–4.47; *p* < 0.001, in PFS and OS, respectively).

### 3.4. Propensity Score-Matching and Inverse Probability Treatment Weighting Analysis

Propensity score-matching (PSM) was used in our studies to reduce the influence of selection bias between the TTFields and non-TTFields groups. The validated cohort is shown in Table 4. The survival risk of the TTFields group was significantly lower than the Non-TTFields group (HR = 0.60; 95% CI, 0.39–0.92; *p* < 0.001 and HR = 0.54; 95% CI, 0.33–0.89; *p* < 0.001, in PFS and OS, respectively) in the propensity score-matching analysis, as shown in Figure 2.

MGMT methylation status impacted the outcomes between the two groups, with elevated risk of non-methylation group (HR = 1.13; 95% CI, 0.39–0.92; *p* < 0.001 and HR = 0.54; 95% CI, 0.33–0.89; *p* < 0.001, in PFS and OS, respectively).

After inverse probability treatment weighting (IPTW), the baseline features of the two groups were balanced, and the significance remained (Table 5).

## 4. Discussion

The phase 3 EF-14 international trial demonstrated the efficacy of TTFields plus TMZ versus TMZ alone as maintenance therapy in patients with newly diagnosed GBM [19]. This led to the approval of TTFields in combination with TMZ for the treatment of newly diagnosed GBM in October 2015. In China, the joint guideline committee of the Chinese Glioma Cooperative Group (CGCG) recently published a guideline and recommended TTFields for the treatment of GBM [23]. While the results of these studies led to approval of FDA and C-FDA for GBM populations, a portion of the neuro-oncology and neurosurgery community remains skeptical of TTFields therapy. The skepticism is due to incoherent results, with certain clinicians taking a wait-it-out approach [24,25]. Here, we retrospectively analyzed the effect of TTFields plus TMZ in newly diagnosed GBM in our center. After observing from 267 GBM patients, TTFields plus TMZ therapy (63 patients) resulted in extended progression-free survival and overall survival compared with temozolomide therapy (204 patients). These findings are consistent with the EF-14 results.

In our current study, the patients of TTFields/TMZ treatment were extracted from the post-marketing registry of newly diagnosed patients in our hospital. The clinical data of TMZ treatment patients were collected from the follow-up library of glioma in Huashan Hospital [26]. There are some apparent differences between the two groups. The TTFields/TMZ group displayed a relatively higher female-to-male ratio, more STUPP scheme, and less MGMT methylation. This reflects the probability that some of the Chinese female patients tended to accept TTFields to prolong survival and agree to shave their heads regardless of aesthetic purposes. In contrast, some Chinese male patients do not want to shave their heads and wear transducer arrays for social reasons. A strength of our study is that measures were taken to reduce those potential biases. We performed a propensity score based on IPTW adjustment, which significantly reduces confounding bias and imbalance in co-variates and thus potentially offers an estimate of treatment effect similar to randomized trials [27,28]. In our preliminary analysis, the median progression-free survival (PFS) in the TTFields/TMZ arm was 16 months (95% CI 9.6–24.6) versus 11 months (95% CI 9–12) with TMZ alone (*p* < 0.05). Median overall survival was 21.8 months (95% CI 17.4-NA) with TTFields/TMZ versus 15 months (95% CI 13–18; HR 0.43, *p* < 0.01) with TMZ group. After adjustment, the arms were well-balanced regarding sex, resection, STUPP scheme, and MGMT promoter methylation. Upon IPTW survival analysis, TTFields/TMZ was associated with a significantly lower risk of death (hazard ratio (HR), 0.19 in OS (95% CI 0.09–0.41) and progression (HR, 0.35 (95% CI 0.14–0.91)) compared with TMZ, which was consistent with preliminary analysis.

In this study, the surgery extension, complete standard Stupp protocol, and TTFields were significantly associated in both univariate and multivariate analyses. In addition, MGMT promoter unmethylation and TERT mutation showed worse survival on univariate analysis. Surgery extension [29], complete standard Stupp protocol [29], and TTFields [29] were previously reported as predictors of improved survival of patients with GBM, consistent with the results of the present study.

Treatment adherence and electric field duration time have emerged as important factors for TTFields efficacy [30,31,32,33]. Patients are recommended to wear the device for at least 18 h per day, with a corresponding adherence rate of ≥75%. While adherence rates were high in both EF-11 and EF-14, the median adherence rate was below the recommended 75% in PRiDe [29]. In our study, median adherence rate was 85%. The median duration of tumor treating fields therapy was longer in our group (10.6 months) compared with that of EF-14 (8.2 months). These findings suggest that high adherence rate and long electric field duration may ensure the effectiveness of electric field therapy in GBM. 

Based on the results of our study, it can be reasonably argued that TTFields should be discussed with Chinese patients who are newly diagnosed with GBM as part of initial therapy. Further studies would be needed to refine the population most likely to benefit, and more importantly, identify subsets where the benefit is minuscule or not present. Future analysis of prolonged and short survival to NovoTTField Therapy will need to include multi-omics analysis of the tumor (exon sequencing, methylation, RNA sequencing, advanced liquid biopsy capacities). We find that there has been important progress in understanding the molecular determinants of glioma invasion and migration, such as growth factors, intracellular signaling cascade, cell–ECM, and cell–cell receptors [29]. Along with ever-improving molecular technologies and their sensitivities, we are hopeful that specific biomarkers involved in glioma invasion and migration will soon be found in tissue, blood, or CSF [34].

In conclusion, our data represent the largest group of patients with newly diagnosed GBM in China, containing 267 patients in total, 63 of whom were treated with NovoTTFields. The results, individually and collectively, demonstrate that adding TTFields to temozolomide chemotherapy can result in statistically significant improvement in progression-free survival and overall survival in Chinese GBM patients.

## Figures and Tables

**Figure 1 jcm-11-05855-f001:**
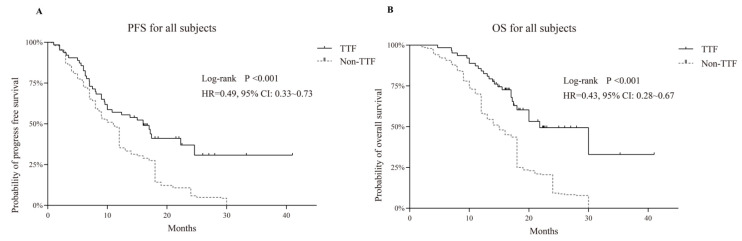
Kaplan–Meier survival analysis in comparing tumor treating fields (TTF) versus non-TTF group; progression-free survival (PFS) is shown in (**A**) and overall survival in (**B**), revealing the prognostic advantages of TTF.

**Figure 2 jcm-11-05855-f002:**
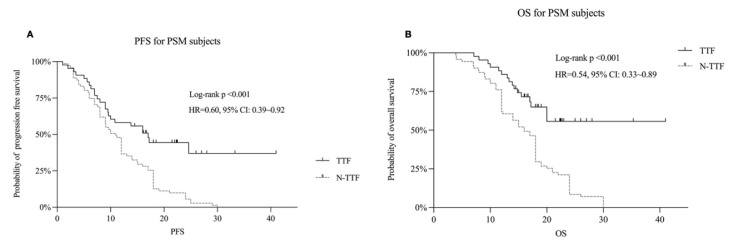
Kaplan–Meier survival analysis after propensity score-matching (PSM) in comparing TTF versus Non-TTF group; progression-free survival (PFS) is shown in (**A**) and overall survival in (**B**).

**Table 1 jcm-11-05855-t001:** Patient characteristics.

		TTF (*n* = 63)	N-TTF (*n* = 204)
Age	Mean ± SD	49.98 ± 13.40	51.72 ± 14.43
	Median	51	54
Sex	Male	30 (48%)	132 (65%)
	Female	33 (52%)	72 (35%)
Baseline KPS	Mean ± SD	80.00 ± 12.05	82.70 ± 14.59
	Median	80.00	90.00
Surgery extension	Gross total resection	44 (70%)	161 (79%)
	Partial resection	10 (16%)	43 (21%)
	Biopsy	9 (14%)	0 (0.00%)
Tumor location	Frontal lobe	18 (29%)	86 (42%)
	Superficial hemisphere	26 (41%)	96 (47%)
	midline/deep structure/infratentorial	19 (30%)	22 (11%)
IDH1-R132H status	Wild type	56 (89%)	180 (88%)
	Mutated	5 (8%)	20 (10%)
	Invalid	2 (3%)	4 (2%)
MGMT promoter region methylation	Methylated	20 (32%)	43 (21%)
	Unmethylated	38 (60%)	66 (32%)
	Invalid	5 (8%)	95 (47%)
TERT promoter mutation status	Unmethylated	23 (37%)	57 (28%)
	Methylated	28 (44%)	52 (25%)
	Invalid	12 (19%)	95 (47%)
Median compliance of TTFields	Mean ± SD	0.85 ± 0.10	
	Median	0.87	
Completed standard Stupp protocol	Yes	63 (100%)	170 (83%)
	No	0 (0%)	34 (17%)

**Table 2 jcm-11-05855-t002:** Univariate and multivariate analyses of progression-free survival (PFS) in GBM patients.

		Univariate Analysis				Multivariate Analysis			
	*n* (%)	HR	95% CI		Crude *p*-Value	HR	95% CI		Adjusted *p*-Value
			Lower	Upper			Lower	Upper	
Age									
<65	215 (81%)	1							
≥65	52 (19%)	1.27	0.93	1.73	0.130				
Sex									
Male	162 (61%)	1							
Female	105 (39%)	0.71	0.55	0.93	0.013	0.84	0.64	1.11	0.219
TTFields usage *									
N-TTF	204 (76%)	1							
TTF	63 (24%)	0.53	0.37	0.75	<0.001	0.49	0.33	0.73	0.001
Baseline KPS									
≤80	82 (31%)	1							
>80	185 (69%)	0.96	0.73	1.27	0.793				
Surgery extension *									
Gross total resection	205 (77%)	1							
Partial resection	53 (20%)	2.17	1.58	2.99	<0.001				
Biopsy	9 (3%)	1.66	0.82	3.39	0.162	2.03	1.56	2.64	<0.001
Tumor location									
Frontal lobe	104 (39%)	1							
Superficial hemisphere ^a^	122 (46%)	1.09	0.83	1.44	0.533				
Midline/deep structure/infratentorial	41 (15%)	1.35	0.92	1.97	0.127				
IDH1-R132H status *									
Wild-type	236 (88%)	1							
Mutated	25 (9%)	0.58	0.37	0.92	0.019				
Invalid	6 (3%)	0.6	0.22	1.61	0.310	0.63	0.44	0.92	0.016
MGMT promoter region methylation									
Methylated	63 (24%)	1							
Unmethylated	104 (39%)	1.54	1.09	2.18	0.014				
Invalid	100 (37%)	1.49	1.06	2.1	0.021	1.09	0.88	1.35	0.443
TERT promoter mutation status									
Unmethylated	80 (30%)	1							
Methylated	80 (30%)	1.58	1.13	2.21	0.007				
Invalid	107 (40%)	1.34	0.99	1.82	0.062	1.03	0.85	1.25	0.786
Completed standard Stupp protocol *									
Yes	233 (87%)	1							
No	34 (13%)	2.19	1.51	3.17	<0.001	1.78	1.21	2.61	0.003

^a^ Frontal lobe was not included. * Demonstrates statistically significance at *p* < 0.05, both univariate and multivariate analyses.

**Table 3 jcm-11-05855-t003:** Univariate and multivariate analyses of overall survival (OS) in GBM patients.

		Univariate Analysis				Multivariate Analysis			
	*n* (%)	HR	95% CI		Crude *p*-Value	HR	95% CI		Adjusted *p*-Value
			Lower	Upper			Lower	Lower	
Age									
<65	215 (81%)	1							
≥65	52 (19%)	1.32	0.96	1.81	0.087				
Sex									
Male	162 (61%)	1							
Female	105 (39%)	0.75	0.57	0.98	0.034	0.82	0.63	1.08	0.154
TTFields usage *									
N-TTF	204 (76%)	1							
TTF	63 (24%)	0.40	0.27	0.60	<0.001	0.43	0.28	0.67	<0.001
Baseline KPS									
≤80	82 (31%)	1							
>80	185 (69%)	1.01	0.76	1.34	0.952				
Surgery extension *									
Gross total resection	205 (77%)	1							
Partial resection	53 (20%)	1.61	1.17	2.22	0.003				
Biopsy	9 (3%)	0.74	0.33	1.68	0.473	1.46	1.12	1.91	0.006
Tumor location									
Frontal lobe	104 (39%)	1							
Superficial hemisphere ^a^	122 (46%)	0.99	0.75	1.31	0.953				
Midline/deep structure/infratentorial	41 (15%)	1.18	0.79	1.75	0.425				
IDH1-R132H status									
Wild-type	236 (88%)	1							
Mutated	25 (9%)	0.81	0.52	1.28	0.373				
Invalid	6 (3%)	0.77	0.29	2.09	0.612				
MGMT promoter region methylation									
Methylated	63 (24%)	1							
Unmethylated	104 (39%)	1.46	1.02	2.08	0.036				
Invalid	100 (37%)	1.57	1.11	2.22	0.010	1.10	0.89	1.35	0.381
TERT promoter mutation status									
Unmethylated	80 (30%)	1							
Methylated	80 (30%)	1.65	1.17	2.32	0.005				
Invalid	107 (40%)	1.57	1.15	2.15	0.005	1.09	0.90	1.32	0.387
Completed standard Stupp protocol *									
Yes	233 (87%)	1							
No	34 (13%)	3.08	2.12	4.47	<0.001	2.53	1.71	3.73	<0.001

^a^ Frontal lobe was not included. * Demonstrates statistically significance at *p* < 0.05, both univariate and multivariate analyses.

**Table 4 jcm-11-05855-t004:** Patient characteristics before and after propensity score-matching (PSM).

	Unmatched Data			Matched Data		
	TTF (*n* = 63)	N-TTF (*n* = 204)	*p*-Value	TTF (*n* = 49)	N-TTF (*n* = 87)	*p*-Value
Age			0.398			0.962
Mean ± SD	49.98 ± 13.40	51.72 ± 14.43		49.41 ± 13.26	49.29 ±14.58	
Sex			0.015			0.445
Male	30 (48%)	132 (64%)		22 (45%)	45 (52%)	
Female	33 (52%)	72 (35%)		27 (55%)	42 (48%)	
Baseline KPS			0.184			0.891
Mean ± SD	80.00 ± 12.05	82.70 ± 14.59		81.84 ± 11.85	82.18 ± 15.21	
Surgery extension			<0.001			0.436
Gross total resection	44 (70%)	161 (79%)		40 (82%)	66 (76%)	
Partial resection	10 (16%)	43 (21%)		9 (18%)	21 (24%)	
Biopsy	9 (14%)	0 (0%)				
Tumor location			<0.001			0.809
Frontal lobe	18 (29%)	86 (42%)		16 (33%)	31 (36%)	
Superficial hemisphere	26 (41%)	96 (47%)		23 (47%)	42 (48%)	
Midline/deep structure/infratentorial	19 (30%)	22 (11%)		10 (20%)	14 (16%)	
IDH1-R132H status			0.780			0.714 *
Wild-type	56 (89%)	180 (88%)	0.744	42 (86%)	73 (84%)	
Mutated	5 (8%)	20 (10%)		5 (10%)	12 (14%)	
Invalid	2 (3%)	4 (2%)		2 (4%)	2 (2%)	
MGMT promoter region methylation			<0.001			0.011
Methylated	20 (32%)	43 (21%)		12 (24%)	32 (37%)	
Unmethylated	38 (60%)	66 (32%)		33 (67%)	36 (41%)	
Invalid	5 (8%)	95 (47%)		4 (9%)	19 (22%)	
TERT promoter mutation status			<0.001			0.43
Unmethylated	23 (37%)	57 (28%)		18 (37%)	38 (44%)	
Methylated	28 (44%)	52 (25%)		23 (47%)	31 (36%)	
Invalid	12 (19%)	95 (47%)		8 (16%)	18 (20%)	
Completed standard Stupp protocol			<0.001			NA
Yes	63 (100.00%)	170 (83%)		49(100%)	87(100%)	
No	0 (0.00%)	34 (17%)		0	0	

* Fisher chi-squared test.

**Table 5 jcm-11-05855-t005:** PFS and OS analysis after inverse probability treatment weighting (IPTW).

PFS					
	HR	Std. Err.	*p*-Value	95% CI	
				Lower	Upper
TTF	0.35	0.17	0.031	0.14	0.91
**OS**					
	**HR**	**Std. Err.**	***p*-Value**	**95% CI**	
				**Lower**	**Upper**
TTF	0.19	0.08	<0.001	0.09	0.41

## Data Availability

The datasets generated during and/or analyzed during the current study are available from the corresponding author on reasonable request.

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
