# Peer review of "Tumor Treating Fields Combine with Temozolomide for Newly Diagnosed Glioblastoma: A Retrospective Analysis of Chinese Patients in a Single Center"

_jcm, 2022, doi:10.3390/jcm11195855_

Round 1

Reviewer 1 Report

The study needs major revision in light of the following comments as it contains many errors that need to be corrected:

1. The abstract section should be reviewed as it has grammatical errors along with the use of unexplained acronym.

2. The introductory section needs to be revised as it has grammatical errors along with the use of misspelled words and punctuation marks which sometimes lead to disruption of the reading flow.

3. In the methods section I could not find the details of the ethics authorization of the study with the complete approval number, consent to the written information provided.

4. The results need a thorough review. Most of tables and figures are free of legends and related descriptions. From Univariate and Multivariate tables, The surgery extension and complete standard Stupp protocol seems to be crucial prognostic factors together with TTFields, why authors did not discuss this aspect? Moreover they wrote that “Gross total resection undisputedly revealed superior prognosis than subtotal resection” but in tables 2 and 3 the gross total resection is set as 1, authors should explain this line. MGMT and TERT status seems to be statistically significant both in unmethylated/methylated and invalid rows, authors should explain why this results is significant. It is unclear where the details regarding the MGMT methylation status after PSM analysis were presented.

5. The Discussion section needs to be revised as it has grammatical errors along with the use of misspelled words and punctuation marks which sometimes lead to disruption of the reading flow.

Author Response

Dear reviewer: 

we really appreciated your opinions, our manuscript and revision were shown in the attachments.  

Reviewer 2 Report

This topic is interesting, but I have some concerns. Look at these points to improve it:

- In the abstract. "These results are consistent with the outcome of EF-14". This is not a conclusion data of your paper. So please remove it.

- "Multimodal therapy of glioblastoma includes surgery, radiotherapy, systemic chemotherapy... " As also second surgery, please consider these refs: -- Impact of recurrence pattern in patients undergoing a second surgery for recurrent glioblastoma. Acta Neurol Belg. 2022 Apr;122(2):441-446. doi: 10.1007/s13760-021-01765-4.  -- Factors impacting survival following second surgery. J Neurooncol. 2014 Mar;117(1):147-52. doi: 10.1007/s11060-014-1366-9.

- "Tumor-treating fields (TTFields) have changed the first-line clinical practice of glioblastoma worldwide by its promising therapeutic efficacy..." So, what does this paper add new to the literature? Please state in the introduction section, improve and report what is the aim of this paper.

- "As female carried with longer PFS and OS", this was also reported in previous paper. In addition 65% of N-TTF group patients are male compared to the other group (48%). Could this lead to some bias on survival? Rebut and discuss.

-  In the discussion section authors wrote: "The clinical data of TMZ treatment patients were collected from the follow-up library of Huashan glioma center[25]". What do authors mean with this sentence? 

- "With ever-improving molecular technologies and their sensitivities, we are hopeful that specific biomarkers will soon be found in tissue, blood, or CSF." This point is not a limitation of the paper and should be moved at the end of discussion section. This point about "tissue, blood, or CSF" should be discussed there. Look also at these refs: --  Therapeutic strategies for inhibiting invasion in glioblastoma. Expert Rev Neurother. 2009 Apr;9(4):519-34. doi: 10.156/442444

- What about the role of gross total resection? 70% Vs 79% in the 2 groups. did authors extract any data?

- add a conclusion section to report what this paper add new to the literature.

Author Response

(The authors gave the same response as above.)

Round 2

Reviewer 1 Report

The authors' word and pdf files contain empty tables 1, 2, 3, 4 and 5.

It is necessary to improve the quality of Figures 1 and 2.

Reviewer 2 Report

good